# REM Sleep: An Unknown Indicator of Sleep Quality

**DOI:** 10.3390/ijerph182412976

**Published:** 2021-12-09

**Authors:** Giuseppe Barbato

**Affiliations:** Department of Psychology, Università degli Studi della Campania Luigi Vanvitelli, 80122 Caserta, Italy; giuseppe.barbato@unicampania.it

**Keywords:** sleep quality, REM sleep, awakening

## Abstract

Standard polysomnographic analysis of sleep has not provided evidence of an objective measure of sleep quality; however, factors such as sleep duration and sleep efficiency are those more consistently associated with the subjective perception of sleep quality. Sleep reduction as currently occurs in our 24/7 society has had a profound impact on sleep quality; the habitual sleep period should fit within what is a limited nighttime window and may not be sufficient to satisfy the whole sleep process; moreover, the use of artificial light during the evening and early night hours can delay and disturb the circadian rhythms, especially affecting REM sleep. The correct phase relationship of the sleep period with the circadian pacemaker is an important factor to guarantee adequate restorative sleep duration and sleep continuity, thus providing the necessary background for a good night’s sleep. Due to the fact that REM sleep is controlled by the circadian clock, it can provide a window-like mechanism that defines the termination of the sleep period when there is still the necessity to complete the sleep process (not only wake-related homeostasis) and to meet the circadian end of sleep timing. An adequate amount of REM sleep appears necessary to guarantee sleep continuity, while periodically activating the brain and preparing it for the return to consciousness.

## 1. Sleep Measures and Sleep Quality

A “good night’s sleep” is a night with a sufficient sleep duration (seven–eight hours for most individuals) that provides sufficient time for the homeostatic restorative process and that is characterized by the generally good “quality” of the whole sleep period. Although the completion of wake–sleep homeostasis appears to be a necessary requisite for a “good night’s sleep”, it is not a factor that per se can define “sleep quality”, which instead refers, as originally suggested by Buysse et al. [1], to “a multifaceted construct that is difficult to characterize by any single correlate”. Unsatisfactory sleep quality is a core feature either in healthy subjects who may complain of non-restorative sleep or in clinical conditions with primary or secondary insomnia.

Sleep diaries and questionnaires such as the Pittsburg Sleep Quality Index [2] are commonly used to evaluate an individual’s quality of night sleep.

Subjective estimates of sleep latency, sleep duration, presence of “insomnia”, use of sleeping drugs and daytime functioning constitute the general background for the assessment of sleep quality; however, none of these measures are able to focus on the specific sleep components that can lie behind sleep quality and they furthermore do not give cues regarding the biological mechanisms that might be responsible for altering the sleep process and disturbing sleep continuity.

Polysomnography (PSG) is the gold standard to objectively evaluate sleep; by measuring sleep latency, sleep duration, sleep efficiency, sleep stages and sleep pattern characteristics, it also provides information about sleep regulation and the possible functions of the two main sleep components individuated throughout PSG: slow-wave sleep (SWS), also known as “delta sleep”, and REM sleep. SWS and REM sleep are controlled by different mechanisms and have different functions [3,4,5]; moreover, these two sleep states occur preferentially during different parts of the sleep period: SWS is mainly located at the beginning of sleep and involves the progressive depletion of the cumulated sleep pressure, whereas REM sleep occurs cyclically and is more often expressed at the end of the sleep period, and it is controlled by short- and long-term homeostatic regulations [6,7,8] and by the circadian clock [9,10].

The present review examines the PSG sleep variables in relation to sleep quality, with a special focus on REM sleep. The critical role of REM sleep within the whole sleep process is discussed.

## 2. Effects of Sleep Durations and Sleep Efficiency on Sleep Quality

Two basic processes are considered to regulate sleep time and duration: a homeostatic process (S) determined by waking and sleep, and a circadian process (C) that defines the alternation of periods with high and low sleep propensity [11,12]. A third, ultradian process, occurring within sleep, regulates the alternation of the two prevalent sleep states, non REM (NREM) sleep and REM sleep [11]. The process S represents the build-up of sleep pressure, which rises with the progression of waking; experimental models of sleep deprivation have shown that slow-wave activity (SWA), the spectral power of SWS, increases with wake prolongation [13]; the cumulated sleep pressure is then depleted during the sleep period and mirrored by the SWA decline. The homeostatic process allows an increase in sleep pressure when sleep is restricted or absent and a reduction in sleep pressure in response to excess sleep. Process C represents the circadian control of sleep occurrence, a clock-like mechanism located in the SCN that defines, during a 24 h period, the alternation of periods with high and low sleep propensity; it is essentially independent of previous waking and sleep, but linked to oscillations of core body temperature [9] and of melatonin secretion [14]. Circadian processes thus contribute to sleep regulation by favoring a decline in arousal at sleep onset and promoting cortisol secretion and REM to terminate sleep.

Total sleep time (TST) appears to be the natural candidate to give a reliable objective estimate of “sleep quality”; however, it should be emphasized that a habitual sleep duration of 7.5 h is achieved only by a limited number of individuals in the general population, with most subjects in our industrialized society sleeping for consistently less than 6 h per night, thus living in a state of chronic sleep restriction; in fact, the duration of 7.5 h appears not sufficient for the completion of sleep’s restorative process. Reduction in the nocturnal sleep periods by as little as 1.3 to 1.5 h for one night results in a reduction in daytime alertness by as much as 32%, as measured by the Multiple Sleep Latency Test (MSLT) [15]. The pressure of external social, leisure and work-related activities undoubtedly is responsible for the curtailment of sleep duration. Studies on sleep extension confirm that humans may be chronically sleep-deprived. Tiller [16] found that older institutionalized persons, who could not sleep for more than 7 h due to institutional schedules, were trained to sleep for up to 9 or 10 h, leading to a significant improvement in their physical condition. Lewis [17] obtained an average value of 7.9 h for men in the Arctic who were permitted to sleep ad lib; the end point of sleep was determined by the sleeper himself, without the use of an external clock. Aserinsky [18] permitted subjects to sleep according to their fullest capacity in a 30 h period, and the amount of sleep obtained ranged from 15 to 27.5 h, with an average of 20 h; “good” sleepers, subjects that obtained the equivalent of at least one night’s sleep during the day, had a higher REM %. Webb and Agnew [19] observed that when subjects were allowed to sleep ad libitum, their sleep time increased by approximately 2 h. Wehr et al. [14] found that subjects placed under the experimental condition of a short photoperiod, with the opportunity to sleep in a darkness window of 14 h for 28 days, compared with the habitual night window of 8 h, greatly increased their sleep duration. Following a sharp rebound of TST during the first nights of the extended dark condition, consistent with what can be considered a recovery from the prior condition of insufficient sleep, multiple nights of sleep were then required to reach a steady-state value for the total sleep duration of approximately 8.5 h. Kamdar et al. [20] allowed fifteen healthy college students reporting minimal daytime sleepiness to sleep as much as possible during a sleep extension period, and extended sleep resulted in a significant increase in MSLT score, with substantial improvements in daytime alertness, reaction time and mood.

In normal subjects, a short duration of sleep causes emotional instability and mood deterioration. Motomura et al. [21] reported that a 5-day period of restricted sleep (4 h) increased the activity of the left amygdala in response to the facial expression of fear. Restricted sleep also resulted in a significant decrease in the functional connectivity between the amygdala and the ventral anterior cingulate cortex (vACC) in proportion to the degree of sleep debt [22]. The sleep extension following self-imposed chronic sleep restriction was instead able to improve mood regulation via prefrontal suppression of amygdala activity [23].

Short sleep duration has also a profound impact on general health, since it has been reported as an important risk factor for hypertension and cardiometabolic disease [24,25], with advice from the influential “American Heart Association” to health organizations to include evidence-based sleep recommendations in their guidelines for optimal health [26]. On the other hand, long sleep duration has also been associated with health issues and an increased risk of mortality [27]. Using meta-regression analyses on a sample of more than 5 million subjects, a linear association between increased mortality and a sleep duration of less than six hours [28] was found, and a linear association between increased mortality, incident cardiovascular disease and longer sleep duration [29] was also found. However, as suggested by Magee et al. [30], the “relationship between sleep duration and mortality could be affected by residual confounding by poor pre-existing health, as reflected by a combination of preexisting illnesses and functional limitations”.

Alteration in sleep duration can also produce significant effects on several metabolic pathways. It has been shown that sleep restriction and/or reduced sleep quality lead to decreased glucose tolerance, without a compensatory increase in insulin secretion, a condition that increases the risk of type 2 diabetes [31,32]. Decreased levels of the anorexigenic hormone leptin and increased levels of the orexigenic hormone ghrelin have been reported after sleep restriction [33]. Stimulation of brain regions sensitive to food stimuli has also been found following sleep restriction, suggesting that sleep loss may lead to obesity through the selection of high-calorie foods [34]. Finally, a clear relationship between sleep reduction and increased body mass index (BMI) has been consistently reported [35,36]. A recent paper [37] looking at the effect of COVID-19-related quarantine on lifestyle and eating habits reported a worsening of sleep quality, particularly in males engaged in smart working, and an increase in BMI.

Akerstedt et al. [38] reported that low subjective sleep quality in the morning was closely related to short total sleep time, low sleep efficiency and many awakenings throughout the night. In a successive paper [39], they investigated the association between TST, sleep stages and sleep continuity measures; sleep efficiency increased with increasing TST, and longer sleep duration was more frequently associated with better subjective sleep quality and was perceived as a “usual” sleep duration and with better quality. Impaired sleep continuity and lower TST also appear to characterize sleep in insomniacs, as shown in a meta-analysis by Baglioni et al. [40]. According to Akerstedt et al. [39], “short TST duration of a recorded sleep in a real-life context may be an indicator of poor objective sleep quality for that particular sleep episode. Because individuals clearly perceived this reduction, it appears that self-reports of poor sleep quality often may be seen as indicators of poor sleep quality”.

Sleep efficiency (SE), the relationship between total sleep time (TST) and total time in bed, is also one of the measurable parameters that can provide an objective estimate of sleep quality. Akerstedt et al. [41] reported that subjective sleep quality was closely related to sleep efficiency, and at least 87% efficiency was required for ratings of “rather good” sleep. Keklund and Akerstedt [42] found slow-wave sleep and sleep efficiency to be the strongest predictors of the sleep quality index (SQI) related to the initiation and maintenance of sleep. Kaplan et al. [43] compared measures obtained by PSG recording with subjective ratings of prior-night sleep quality, and objective sleep efficiency emerged as the strongest correlate of subjective sleep quality across all models, and across both sexes. Greater total sleep time and sleep stage transitions were also significant. According to their findings, the age of the subject contributed to the relationship between sleep efficiency and sleep quality, with the oldest adults reporting the highest sleep quality even as objective sleep deteriorated.

Sleep microstructure can also provide information on sleep continuity and stability. Laffan et al. [44] analyzed the polysomnography recordings of a sample of 5684 participants in the “Sleep Heart Health Study”; a high overall sleep stage transition rate was associated with restless and light sleep fragmentation, and an analysis of stage-specific transition rates showed that transitions between wake and NREM sleep were also independently associated with restless and light sleep. Further analysis of the sleep microstructure is provided by the measurement of the cyclic alternating pattern (CAP) of NREM sleep [45]. The CAP rate is considered to reflect the arousal level during sleep. Terzano et al. [46] reported that the altered CAP rate in subjects with insomnia was reduced by hypnotic treatment and that the most significant correlation between sleep quality and PSG variables was found for the CAP rate.

Analysis of sleep stages in different age groups has suggested possible associations with subjective sleep quality for an increased duration of stage 2 sleep [47,48], and for an increased duration of slow-wave sleep [48].

Limited evidence on objective indicators of sleep quality has emerged from studies on spectral power EEG. Gabryelska et al. [49] have shown that higher subjective sleep quality was related to decreased NREM stage 2 sigma 2 and REM delta 1; however, very small effect sizes of correlations between sleep quality and spectral power were found, suggesting, as stated by the authors, that “the amount of variance in subjective sleep quality that can be explained trough EEG power spectral density is small”. In contrast with the previous literature, they also reported a negative correlation with total sleep duration, NREM stage 2 and REM sleep.

## 3. Sleep Quality, SWS and Daytime Functioning

Since the main sleep function is to allow wake functioning, a measure of sleep quality can also refer to the ability to stay fully awake and adequately perform during the day. It should be considered that sleep quality can influence not only daytime functioning but can significantly affect cognitive abilities and mood, making it a critical factor to guarantee mental health.

According to the suggested role of sleep to provide recovery from the previous wake period, SWS is considered the main sleep component to satisfy homeostatic needs and consequently favor good cognitive performance during the following day.

The role of SWS has been mainly investigated using models designed to deprive participants of this sleep stage. Selective SWS deprivation protocols, which disrupt SWS occurrence through acoustic stimulation or frequent awakenings, have shown a significant rebound of SWS in the subsequent night [50,51], suggesting that increased SWS pressure follows its deprivation. Increases in daytime sleep propensity following SWS deprivation have been also reported [52,53]. Furthermore, experimental protocols that have indagated the effect of sleep deprivation on recovery sleep suggest that slow-wave sleep has priority when sleep is recovered following sleep deprivation [54].

Although recent studies have suggested that slow-wave sleep is the key component for sleep functions, i.e., consolidation of learning material experienced during previous wake period [55] and restoration of synaptic homeostasis [56], its significance for waking performance and its role as a fundamental component of sleep quality have received no consistent support. Keklund and Akerstedt [42], using measures derived from the Karolinska Sleep Diary, found that slow-wave sleep (SWS) and sleep efficiency were the strongest predictors of the sleep quality index (SQI), which refers to the initiation and maintenance of sleep, suggesting that sleep quality seems to be related to the depth of sleep and sleep continuity. Akerstedt et al. [38] did not find any relation between sleep quality ratings and SWS. Kaplan et al. [43] found that the amount of slow-wave sleep was not determined to be important in predicting subjective sleep quality.

SWS deprivation appears to not consistently affect subsequent waking performance. Agnew et al. [54,57], in two studies on five subjects for two consecutive nights and on five subjects for seven consecutive nights, systematically deprived subjects of stage 4 sleep, using either tone stimuli or mild electric stimulation during EEG occurrence of SWS. Both studies reduced stage 4 and a rebound of stage 4 occurred in recovery sleep; stage 4 deprivation produced a depressed outlook among subjects, with a reported increase in negative somatic feelings, no effect on tests of reaction time and addition tests administered on waking were found. Bonnet [58] did not show differences in cognitive performance, mood or nap latency measure between a sleep condition with a preserved amount of SWS and a sleep condition with no SWS (subjects aroused whenever they entered stage 3 sleep). Gillberg and Akerstedt [53] evaluated the effects of sleep curtailment and SWS suppression on daytime alertness. Seven subjects participated in four conditions: undisturbed 8 h sleep, undisturbed 4 h sleep (4U; 03.00–07.00 h), 4 h sleep (4D; 03.00–07.00 h) that was acoustically disturbed when delta waves appeared and a condition with no sleep (0). Reaction time performance was significantly better during the undisturbed 8 h sleep; however, sleep duration was more important for daytime alertness than SWS content. Van Der Werf et al. [59] used automated electroencephalogram (EEG) dependent acoustic feedback aimed at selective interference with—and reduction of—SWA. The number of vigilance lapses increased as a result of SWA reduction, irrespective of the type of vigilance task. Recognition on the declarative memory task was also affected by SWA reduction; however, this did not affect reaction time on either of the vigilance tasks or implicit memory task performance.

## 4. Sleep Quality and REM Sleep

Despite its pioneering role in the scientific approach to sleep and its fascinating nature due to the link with dream activity, REM sleep is not generally considered to have a fundamental sleep function, especially when looking at homeostasis and wake performance. Studies with antidepressant drugs that significantly reduce REM sleep [60] have also shown no significant impairment in cognitive abilities; instead, according to Vogel et al. [61], suppression of REM sleep is considered a possible mechanism for antidepressant activity. On the other hand, Leary et al. [62] have recently shown that a decreased percentage of REM sleep is associated with a greater risk of all-cause, cardiovascular and other noncancer-related mortality in middle-aged and older adults.

Motomura et al. [22] reported that reduced connectivity between the medial prefrontal cortex and amygdala, which is involved in mood deterioration under sleep deprivation, correlated with REM reduction, suggesting that adequate REM sleep may be important for mental health maintenance.

As previously discussed in the present paper, among the different PSG variables suggested as objective indicators of sleep quality, short total duration of sleep and decreased sleep efficiency appear to be those more consistently reported as indicators of low sleep quality. Considering that a short sleep duration implicates a significant reduction in REM sleep, since REM sleep mostly occurs during the second half of the night, this sleep component can be critical in determining sleep quality; alteration of the systems controlling REM sleep can also produce sleep fragmentation, contributing to decreased sleep efficiency. Furthermore, the use of artificial light during the evening and early nighttime hours can delay and disturb circadian rhythms, especially affecting REM sleep. Chang et al. [63] found that, compared with reading a printed book in reflected light, reading an LE eBook in the hours before bedtime delayed the phase of the endogenous circadian pacemaker that drives the timing of the daily rhythms of melatonin secretion, sleep propensity and REM sleep propensity and impaired morning alertness. Subjects in the LE eBook condition had significantly less REM sleep and were sleepier in the morning. The phase delay of the circadian rhythm reduces the length of the sleep period, thus limiting the occurrence of REM sleep, which is normally concentrated in the last part of the sleep period.

In the seminal work of Webb and Agnew [64] that sought to define natural sleep duration, long sleepers compared to short sleepers showed more REM sleep with no evidence of increased SWS; short sleepers, however, showed a higher percentage of SWS, suggesting that the longer sleep period was achieved by increasing the REM duration. Similarly, Aeschbach et al. [65] reported equal amounts of slow-wave sleep in short and long sleepers, with a higher percentage of SWS in short sleepers, whereas, in long sleepers, REM was both increased in duration and higher in percentage.

An increased duration of REM sleep has been also reported together with an increased duration of total sleep in extended sleep during an LD (light/dark) 10:14 photoperiod study [14,66]; furthermore, compared to the habitual LD 16:8 (light/dark) condition, subjects in this “winter” photoperiod reported increased vigor and a diminished level of fatigue, according to their Profile of Mood States and 100 mm line rating scale [14]. Akerstedt et al. [39] have reported in a large population study that sleep efficiency, NREM stage 1 and stage 2, REM duration and REM % increased with increased total sleep duration; long sleep periods were also perceived to be of better quality. As occurred for sleep continuity, REM sleep duration strongly increased with sleep duration. Although longer REM sleep may simply reflect the longer duration of the sleep period, in the Akerstedt et al. [39] study, also the relative amount of REM (%) increased with increasing sleep duration, suggesting, as stated by the authors, that “may also reflect a sensitivity of REM % to poor sleep continuity as REM % is reduced in insomniacs”.

Riemann et al. [67] have proposed that, in insomnia, “instability” of REM sleep contributes to the experience of disrupted and non-restorative sleep, providing a possible explanation for the discrepancy between minor objective alterations in standard parameters of sleep continuity and the profound subjective impairments in these patients.

Della Monica et al. [68] have shown that self-reported sleep quality is positively associated with the duration of REM sleep, with no significant association with slow-wave sleep. Furthermore, an analysis of performance measures assessed with the goal neglect task showed that fewer awakenings and more REM sleep were associated with better executive function. A significant effect of REM sleep on wake abilities has been previously reported by Feinberg et al. [69], who found a positive association between REM measures (REM duration and REM activity) and cognitive performance as assessed by the Wechsler Adult Intelligence Scale and the Wechsler Memory Scale. REM sleep was also shown to be associated with less cognitive decline in longitudinal studies conducted in elderly subjects. Using data from the (MrOS) Study, Song et al. [70] found that, after adjusting for 15 potentially confounding variables (e.g., age, depression, hypertension), men with the lowest quartile of REM sleep (<15%) and the highest quartile of N1 sleep (≥8.52%) showed an accelerated rate of cognitive decline, as assessed by the Trails B Test and a Modified Mini-Mental State Examination. Pase et al. [71] showed that more REM sleep, but not SWS, was protective against the emergence of dementia.

The fundamental role of REM sleep in the memory process has been further highlighted by experimental models in animals; using a combination of electro-physiological recording and optogenetic techniques, Boyce et al. [72] have demonstrated that neural activity occurring specifically during REM sleep is required for spatial and contextual memory consolidation.

Physiological consequences of REM loss have also been reported, including inflammation and interleukin-17 elevation [73], alteration of the immune system [74] and increased sensitivity to pain [75]. All of these can significantly contribute to impaired sleep quality.

## 5. Sleep Quality and REM Density

REM density, the frequency of eye movements during a REM period, is another REM measure that can have predictive value regarding sleep quality. REM density appears to be regulated by different mechanisms to those that control the duration of REM sleep. Feinberg et al. [76] proposed that REM density may be related to the level of arousal, with REM density being lower when sleep is deeper. They observed that sleep deprivation increased the level of slow-wave activity, which is an indicator of sleep pressure, and reduced REM density in recovery sleep. REM density typically increases across the night as sleep pressure progressively diminishes. Barbato et al. [77] reported that the propensity to wake from sleep is higher in the REM sleep period with a high density of REM than from the NREM sleep period, possibly reflecting an increased level of the brain arousal process associated with REM sleep. Increased REM density has been reported in depression [78] and post-traumatic stress disorder [79], both conditions characterized by hyperarousal. Recent findings have identified “restless REM”, a condition of fragmented REM sleep with a high frequency of eye movements, as an important marker of insomnia [80]; furthermore, in this patient group, higher REM density together with REM arousal was strongly associated with the slow dissolution of emotional distress.

Aserinsky [18] found that, in extended sleep, REM density increased with each successive REM episode, approaching a maximum value after 7.5–10 h of sleep; thereafter, periods of wakefulness alternated with periods of sleep, with no further changes in REM density levels. According to Aserinsky [18], the levelling-off of REM density as the night proceeds could reflect the satisfaction of a sleep need, or the build-up of a pressure to awaken, and thus may serve as an index of sleep satiety. As elegantly suggested by Dijk and von Schantz [81], “during a nocturnal sleep episode, the sleep-dependent dissipation of sleep propensity is counteracted by a circadian increase in sleep propensity, thereby facilitating sleep consolidation until the very end of the habitual sleep episode. Subsequently, a gate to wakefulness is created by the combined circadian and sleep-dependent promotion of REM sleep and REM density at the appropriate time”.

Alteration of the frequency of REM has been reported in relation to cognitive performance. Feinberg et al. [69] first showed that low REM activity in healthy older adults was associated with lower performance on psychometric tests. Spiegel et al. [82] reported that low REM density predicted the level of cognitive decline in older subjects. In the already cited work of Akerstedt et al. [39], who reported a positive association between sleep quality and total sleep duration, REM intensity (the amplitude of rapid eye movements) decreased markedly with increased TST, further suggesting that REM measurement can provide an index of sleep quality.

## 6. Why REM Sleep Could Be a Sensitive Indicator of Sleep Quality

The prevailing assumption that slow-wave sleep is the core sleep component has probably contributed to the notion that REM sleep has a secondary but not essential role; however, as occurs for habitual sleep, which might not be of sufficient duration, healthy individuals can also be REM-sleep-deprived. Scorucak et al. [83] have shown that seven days of sleep restriction (6 h time in bed) resulted primarily in a reduction in REM sleep; interestingly, during the following sleep extension, the REM sleep duration increased. As stated by Scorucack et al., “in the search for the mechanisms underlying the negative consequences of insufficient sleep, the implication of a REM sleep deficit should be considered”. Klerman et al. [84] have recently reported a striking increase in REM sleep during sleep extension over baseline, suggesting a rebound phenomenon from a possible REM deprivation that occurs in a habitual sleep opportunity. A rebound in REM sleep has been also observed after a space shuttle mission, during which sleep duration was approximately 6.5 h [85]. Singh et al. [86] reported the occurrence of SOREMP (short REM latency period), an index of increased REM pressure, in a population sample of 333 adults. Of the variables assessed (MSLT, Epworth Sleepiness Scale and total sleep time from nocturnal polysomnography), objective sleepiness, as determined by the MSLT, was the only measure significantly associated with two or more SOREMPs. 

Whereas slow-wave sleep might serve a need created by waking, the functional significance of REM sleep is less clear. Benington and Heller [87] have suggested a homeostatic relationship between NREM sleep and REM sleep, where REM serves a need created during NREM sleep. Data in animals [87,88] and in humans [66] have shown that the REM intervals are regulated by homeostatic rules. REM pressure accumulates during the NREM episodes and is dissipated during the successive REM episode. A longer REM episode causes stronger dissipation of this pressure and thus requires a longer interval to accumulate pressure before the next REM episode can occur. Disruption of REM sleep, by interrupting the pressure dissipation, can impair the quality and continuity of NREM sleep, since, due to REM’s homeostatic mechanism, interruption of REM discharge results in more frequent attempts to enter REM, causing the fragmentation of NREM sleep [89].

Most hypotheses concerning REM sleep have focused on the similarity between neurophysiological events occurring in REM sleep and wakefulness [90]. Broughton [91] reported that the visual evoked potentials of subjects after a forced arousal from REM sleep were similar to those of wakefulness, whereas, after a forced arousal from stage 4 (SWS), subjects showed slower visual evoked potentials. Langford et al. [92] showed that spontaneous arousal from sleep was more likely to occur in REM sleep than in other sleep stages. Lavie et al. [93] found that subjects instructed to wake at a specified time during the night, with no aid of an external alarm clock, mainly awakened from REM sleep. As originally hypothesized by Moruzzi [94], a continuum of increasing arousal levels exists throughout NREM sleep, REM sleep and waking.

A possible REM sleep function is to provide the periodic activation of the brain during sleep, without inducing wakefulness or disturbing the continuity of sleep [95]. Klemm [96] has proposed that the brain uses REM to help wake itself up after it has had a sufficient amount of sleep. Similarly, Horne [97] has suggested that REM seems more likely to prepare for ensuing wakefulness than providing recovery from prior wakefulness, as happens with “deeper” NREM.

The correct phase relationship of the sleep period with the circadian pacemaker is also an important factor to guarantee adequate restorative sleep duration and sleep continuity, thus providing the necessary background for a good night’s sleep. Due to the fact that REM sleep is controlled by the circadian clock [98], it can provide a window-like mechanism that defines the termination of the sleep period when there is still the necessity to complete the sleep process (not only wake-related homeostasis), and to meet the circadian end of sleep timing. Furthermore, the cyclic occurrence of REM sleep could contribute to sleep continuity, while periodically activating the brain and preparing it for the return to consciousness.

Mechanisms that underlie REM occurrence during the night appear thus functional to guarantee sleep continuity and duration.

Arousal processes “monitored” by the frequency of REM can finally further delineate the end of sleep and the awakening, consistently with the reported evidence that awakenings occur much more likely out of a REM period characterized by a high frequency of REM [99,100]. REM sleep can thus be necessary to guarantee sleep continuity, prepare the brain for the wake period and define the natural end of the sleep process.

## 7. Conclusions

Standard PSG parameters and quantitative EEG measures have provided no definitive evidence of a possible objective indicator of sleep quality in healthy individuals [101,102]. Moreover, the manipulation of sleep, either as sleep restriction or deprivation of a specific sleep component (i.e., SWS), has failed to reveal significant and reproducible effects on subsequent waking performance; however, factors such as sleep duration and sleep efficiency are those more consistently associated with the subjective perception of sleep quality.

PSG has been mainly used for traditional sleep scoring analysis, with a few studies addressing the analysis of sleep stage transitions and of cyclic alternating patterns, which, in patient populations, have shown promising insights into the mechanisms of sleep organization. Future studies on sleep quality in healthy subjects should thus be extended to the analysis of sleep microstructure, to better characterize those mechanisms of sleep that can result in poor sleep quality. In addition, PSG assessment of sleepiness and wakefulness with MSLT and MWT, which are used at clinical level to diagnose sleep disorders, will contribute to providing objective information on the daily functioning of healthy subjects.

One aspect that clearly emerges from the available literature is that sleep reduction as currently occurs in our 24/7 society has a profound impact on sleep quality, and artificial light at night (LAN) has extended the average length of the day and delayed the secretion of melatonin; thus, the habitual sleep period has to fit within what is a limited nighttime window and may not be sufficient to satisfy the whole sleep process, especially sacrificing the last part of the sleep period, which contains the larger quota of REM sleep.

As emphasized in a recent paper by Nayman [103], we are experiencing “a silent epidemic of REM sleep deprivation”. Future studies should further clarify the negative consequences of REM sleep deficit and its role in providing a “good night’s sleep”.

## Data Availability

Not applicable.

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
