# Peer review of "REM Sleep: An Unknown Indicator of Sleep Quality"

_ijerph, 2021, doi:10.3390/ijerph182412976_

Round 1

Reviewer 1 Report

I read with interest the manuscript, "REM sleep: an unknown indicator of sleep quality." Personally, I believe it is an interesting issue to analyze and investigate. A lot of thought was put into the writing of this review. I believe that some fundamental facts and questions should be put in the article to make it more complete and comprehensive.

1. When it reads secondary or primary insomnia in line 31, it should be primary or secondary insomnia, correct?

2. Abbreviation for Total sleep duration should be changed to TSD in line 52, or if the author wishes to preserve TST, the abbreviation for Total sleep time should be used instead of duration in the same line.

3. Line 94, it should read pre-existing health concerns instead of preexisting health.

4. Abbreviations are not appropriately utilized, for example, SWS should be used instead of slow wave sleep in line 138 and 154. Throughout the article, be sure to address these concerns.

5. The lines 158-159 should be put in the same sentence as the preceding line, or they might be divided into a distinct phrase.

6. In lines 160-161, the author should provide the model system that was utilized in the experiments in order to make it more significant. This may be used in various sections of the document to make it more informative.

7. The physiological repercussions of rapid eye movement (REM) sleep are not addressed in the text. REM loss has a deleterious influence on the immune system and causes oxidative stress and cell death in animal models, which has been well documented but overlooked in this manuscript. 

8. It is not covered at any length in this article the genes, molecular markers, and other clinical indicators that are connected with poor sleep quality. I hope this lies in the area of hypothesis or target of this manuscript. 

9. In the conclusion section, it is necessary to emphasize the present relevance and future concerns of the relationship of REM sleep with sleep quality assessment and its significance.

10. Sleep quality measurement is the most difficult task since it is very subjective in most cases. The question is, how can we settle this? Are there any suggestions that the author would want to make to the broader public along these lines?

Author Response

I read with interest the manuscript, "REM sleep: an unknown indicator of sleep quality." Personally, I believe it is an interesting issue to analyze and investigate. A lot of thought was put into the writing of this review. I believe that some fundamental facts and questions should be put in the article to make it more complete and comprehensive.

  1. When it reads secondary or primary insomnia in line 31, it should be primary or secondary insomnia, correct?

Yes, it has been corrected

  1. Abbreviation for Total sleep duration should be changed to TSD in line 52, or if the author wishes to preserve TST, the abbreviation for Total sleep time should be used instead of duration in the same line.

Yes, total sleep time (TST) has been used and time used in the same line

  1. Line 94, it should read pre-existing health concerns instead of preexisting health.

Yes, the typo has been corrected

  1. Abbreviations are not appropriately utilized, for example, SWS should be used instead of slow wave sleep in line 138 and 154. Throughout the article, be sure to address these concerns.

Yes, SWS has been used

  1. The lines 158-159 should be put in the same sentence as the preceding line, or they might be divided into a distinct phrase.

Yes, the two sentences have been divided as distinct phrase

  1. In lines 160-161, the author should provide the model system that was utilized in the experiments in order to make it more significant. This may be used in various sections of the document to make it more informative.

The model used in the study was reported, as well as information relative to other studies were added when available, lines 209-230

Agnew et al. [57][54], in two studies on respectively 5 subjects for 2 consecutive nights and on 5 subjects for 7 consecutive nights, systematically deprived subjects of stage 4 sleep, using either a tone stimuli or mild electric stimulation during EEG occurrence of SWS. Both studies reduced stage 4 and a rebound of stage 4 occurred in recovery sleep, stage 4 deprivation produced a depressed outlook on subjects, with a reported increase in negative somatic feelings, no effect on tests of reaction time and addition tests administered on waking were found. Bonnet [58] did not show differences in cognitive performance, mood, or nap latency measure between sleep condition with preserved amount of SWS and sleep condition with No SWS (Subjects aroused whenever they entered stage 3 sleep). Gillberg and Akerstedt [53] evaluated the effects of sleep curtailment and SWS-suppression on daytime alertness. Seven subjects participated in four conditions: an undisturbed 8-h sleep, an undisturbed 4-h sleep (4U; 03.00-07.00 hours), a 4-h sleep (4D; 03.00-07.00 hours) that was acoustically disturbed when delta waves appeared, and a condition with no sleep (0). Reaction time performance was significantly better during the undisturbed 8 h sleep, however sleep duration resulted more important for daytime alertness than SWS content. Van Der Werf et al. [59], used automated electroencephalogram (EEG) dependent acoustic feedback aimed at selective interference with-and reduction of-SWA. The number of vigilance lapses increased as a result of SWA reduction, irrespective of the type of vigilance task. Recognition on the declarative memory task was also affected by SWA reduction, that however did not affect reaction time on either of the vigilance tasks or implicit memory task performance.

  1. The physiological repercussions of rapid eye movement (REM) sleep are not addressed in the text. REM loss has a deleterious influence on the immune system and causes oxidative stress and cell death in animal models, which has been well documented but overlooked in this manuscript. 

The following paragraph has been added to the revised paper, lines 305-307

Physiological consequences of REM loss have also been reported for inflammation and interleukin-17 elevation [73], alteration of immune system [74], and increased sensitivity to pain [75]. All of these can significantly contribute to impair sleep quality.

  1. It is not covered at any length in this article the genes, molecular markers, and other clinical indicators that are connected with poor sleep quality. I hope this lies in the area of hypothesis or target of this manuscript. 

The following paragraph has been added to the revised paper, lines 124-135

Alteration in sleep duration can also produce significant effects on several metabolic pathways. It has been shown that sleep restriction and/or reduced sleep quality lead to decreased glucose tolerance, without compensatory increase in insulin secretion, a condition that increases the risk of type 2 diabetes [31] [32]. Decreased levels of the anorexigenic hormone, leptin and increased levels of the orexigenic hormone, ghrelin, have been reported after sleep restriction [33]. Stimulation of brain regions sensitive to food stimuli has also been found following sleep restriction, suggesting that sleep loss may lead to obesity through the selection of high-calorie food [34]. Finally, a clear relationship between sleep reduction and increased body mass index (BMI) has been consistently reported [35] [36]. A recent paper [37] looking at the effect of COVID-19-related quarantine on lifestyle and eating habits, reported a worsening of sleep quality, particularly in males doing smart working, and an increase of BMI.

  1. In the conclusion section, it is necessary to emphasize the present relevance and future concerns of the relationship of REM sleep with sleep quality assessment and its significance.

The conclusion has been modified, lines 423-432

One aspect that clearly emerges from the available literature is that sleep reduction as currently occur in our 24/7 society has a profound impact on sleep quality, artificial light at night (LAN) has extended the average length of day and delayed the secretion of melatonin, thus the habitual sleep period should fit in what is a limited night window and may not be sufficient to satisfy the whole sleep processes, especially sacrificing the last part of the sleep period which contain the larger quota of REM sleep.

As emphasized in a recent paper by Nayman [104], we are experiencing “a silent epidemic of REM sleep deprivation”, future studies should further clarify the negative consequences of REM sleep deficit and its role in providing a “good night sleep.

  1. Sleep quality measurement is the most difficult task since it is very subjective in most cases. The question is, how can we settle this? Are there any suggestions that the author would want to make to the broader public along these lines?

The following paragraph has been added, lines 414-422

PSG has been mainly used for traditional sleep scoring analysis, with few studies addressed to analysis of sleep stage transitions and of cyclic alternating pattern, that in patient populations have shown promising insights on the mechanisms of sleep organization. Future studies on sleep quality in healthy subjects should thus be extended to analysis of sleep micro structure, to better characterize those mechanisms of sleep that can result in a poor sleep quality. Also, PSG assessment of sleepiness and wakefulness with MSLT and MWT, that are used at clinical level to diagnose sleep disorders, will contribute to provide objective information on daily functioning of healthy subjects.

Reviewer 2 Report

This paper should have a different title which relates to REM sleep, circadian rhythms and sleep quality since these are it’s topics.

The first chapter could benefit from subtitles i.e. effects of sleep length (on sleep quality), factors influencing sleep duration etc.

The second chapter „Sleep quality, slow wave sleep and daytime funcitoning“ should also refer to methods as MSLT and MWT as parameters of sleepiness and wakefulness and not only to vigilance. The function of REM sleep for cognition is barely touched and certainly is another important parameter for sleep quality and circadian rhythm.

The chapter „sleep quality and REM sleep“ should definitely include a few statements about the two process model and the integration of REM sleep into it, as well as some explanations about basic human circadian rhyhtm.

The chapter „conclusion“ finally is one of the essential chapters explaining the role of REM sleep within the homeostatic process. This should be an extra chapter.

The whole article needs some more introduction into circadian rhythms, sleep and wake regulation and should be much more structured.

Author Response

This paper should have a different title which relates to REM sleep, circadian rhythms and sleep quality since these are it’s topics.

I appreciate the reviewer’s suggestion, however the idea of the paper is to emphasize the role of REM sleep in sleep quality, a statement has been added in the revised paper, line 52-54

The present review examines the PSG sleep variables in relation to sleep quality, with a special focus on REM sleep. The critical role of REM sleep within the whole sleep process is discussed.

The first chapter could benefit from subtitles i.e. effects of sleep length (on sleep quality), factors influencing sleep duration etc.

According to the reviewer’s suggestion, the first chapter has been divided in two section, the second section title. Line 56

Effects of sleep durations and sleep efficiency on sleep quality

The second chapter „Sleep quality, slow wave sleep and daytime functioning“ should also refer to methods as MSLT and MWT as parameters of sleepiness and wakefulness and not only to vigilance.

The following paragraph have been added:

 lines 79-81; lines 420-422

Reduction of the nocturnal sleep periods by as little as 1.3 to 1.5 hours for 1 night result in reduction of daytime alertness by as much as 32% as measured by the Multiple Sleep Latency Test (MSLT) [15].

Also, PSG assessment of sleepiness and wakefulness with MSLT and MWT, that are used at clinical level to diagnose sleep disorders, will contribute to provide objective information on daily functioning of healthy subjects.

The function of REM sleep for cognition is barely touched and certainly is another important parameter for sleep quality and circadian rhythm.

The REM role for cognition is discussed in different paragraphs, lines 285-299:  336-343; the following sentence has been also added, lines 300- 304

The fundamental role of REM sleep in memory process has been further highlighted by experimental models in animals, using a combination of electro-physiological recording and optogenetic techniques, Boyce et al. [72] have demonstrated that neural activity occurring specifically during REM sleep is required for spatial and contextual memory consolidation.

The chapter “sleep quality and REM sleep“ should definitely include a few statements about the two process model and the integration of REM sleep into it, as well as some explanations about basic human circadian rhythm.

According to reviewer’s suggestion two process model and circadian rhythm have been added, lines 58-73

Two basic processes are considered to regulate sleep time and duration: a homeostatic process (S) determined by waking and sleep, and a circadian process (C) that defines alternation of periods with high and low sleep propensity [11][12]. A third, ultradian process, occurring within sleep, regulates the alternation of the two prevalent sleep states, nonREM (NREM) sleep and REM sleep [11]. The process S represents the build-up of sleep pressure which rises with the progression of waking, experimental models of sleep deprivation have shown that slow wave activity (SWA), the spectral power of SWS, increases with wake prolongation [13], the cumulated sleep pressure is then depleted during the sleep period and mirrored by the SWA decline. The homeostatic process allows to increase sleep pressure when sleep is restricted or absent and to reduce sleep pressure in response to excess sleep. Process C represents the circadian control of sleep occurrence, a clock-like mechanism located in the SCN defining during the 24 hour period the alternation of periods with high and low sleep propensity, and being basically independent from previous waking and sleep, but linked to oscillations of core body temperature [ 9] and of melatonin secretion [14].Circadian processes thus contribute to sleep regulation by favoring decline of arousal at sleep onset and promoting cortisol secretion and REM to terminate sleep.

The chapter „conclusion“ finally is one of the essential chapters explaining the role of REM sleep within the homeostatic process. This should be an extra chapter.

According to reviewer’s suggestions, the conclusion chapter has been divided, a chapter is dedicated to REM sleep, line 345

Why REM sleep could be a sensitive indicator of sleep quality

The whole article needs some more introduction into circadian rhythms, sleep and wake regulation and should be much more structured.

The revised paper has been changed adding information on circadian rhythm and sleep wake regulation, introduction and conclusion have also been changed, with according to the reviewer’s suggestion, specific chapter for sleep duration and significance of REM

Reviewer 3 Report

Manuscript ID: IJERPH – ijerph-1470006

Title: " REM sleep: an unknown indicator of sleep quality "

The aim of this mini review is to reconsider the role of REM sleep as possible marker of sleep quality.

The topic of the manuscript is potentially interesting. However, I have some concerns on the present form.

Major concerns

In the present form, the manuscript seems linked to the old dichotomy NREM/REM. As the same Author underlines in the Introduction section, sleep quality is a construct linked to several factors that is difficult to characterize through a single correlate. Author should better explain in which framework he inserts the role of REM sleep and which relevance could have in moderating sleep quality.

There are two categories of sleep quality markers: a first type that refers to a specific feature of sleep (as sleep onset latency or Slow Wave Sleep); a second type that derives from a combination of more features (as sleep efficiency or Total Sleep Time). Such categorization does not clearly emerge in the present form of the manuscript. Precisely, because the concept of sleep quality is multifactorial, the indicators of the first type have proved unreliable. On the contrary, the indicators of the second type are more effective from an operational point of view. Reconsidering the possible role of REM sleep as parameter to assess sleep quality could seem unsuccessful if we consider it as an indicator of the first type, unless this reassessment is inserted within a more articulated framework.

Another somewhat confused element is the lack of distinction between the macro and micro structure of sleep. It seems to me that adopting this distinction could improve the work. In this perspective, it should be very important a reference to the Cyclic Alternating Pattern (CAP), completely missing in this version.

I do not entirely agree with the statement that the subjective measures are not able to focus on the sleep features that can guarantee sleep quality. Perhaps, the author should better explain what he means by sleep quality.

Minor concerns

When Author quotes works on sleep ad libitum, he should comment this taking into account the spontaneous trend of biological clock to run longer than 24 hours.

At page 2, line 96, it is present one repetition mistake: “low subjective sleep quality”.

At page 3, line 152, a mark should be moved.

Out of 76 bibliographic references, only 5 (6.5%) are from the last two years.

In the references section, there are some words in bold style that should be changed (for example, in the reference number 32 the word “spectral”).

Author Response

The aim of this mini review is to reconsider the role of REM sleep as possible marker of sleep quality.The topic of the manuscript is potentially interesting. However, I have some concerns on the present form.

 Major concerns

 In the present form, the manuscript seems linked to the old dichotomy NREM/REM. As the same Author underlines in the Introduction section, sleep quality is a construct linked to several factors that is difficult to characterize through a single correlate. Author should better explain in which framework he inserts the role of REM sleep and which relevance could have in moderating sleep quality.

The role of REM sleep for sleep quality is introduced and discussed on lines 52-53; 245-253;

The present review examines the PSG sleep variables in relation to sleep quality, with a special focus on REM sleep. The critical role of REM sleep within the whole sleep process is discussed.

As previously discussed in the present paper, among the different PSG variables suggested as objective indicators of sleep quality, short total duration of sleep and decreased sleep efficiency appear to be those more consistently reported as indicators of low sleep quality. Considering that short sleep duration implicates a significant reduction of REM sleep, since REM sleep mostly occur during the second half of the night, this sleep component can be critical in determining sleep quality, alteration of the systems controlling REM sleep can also produce sleep fragmentation contributing to a decreased sleep efficiency. Furthermore, use of artificial light at evening and early night hours can delay and disturb circadian rhythms, especially affecting REM sleep

In this revised version a whole chapter is specifically dedicated to discuss the role of REM sleep, lines 345-407

Why REM sleep could be a sensitive indicator of sleep quality

The prevailing assumption that slow wave sleep is the core sleep component has probably contributed to assign to REM sleep a secondary not essential role, however as occur for habitual sleep, that might not be of sufficient duration, healthy individuals can also be REM sleep deprived. Scorucak et al. [84] have shown that seven days of sleep restriction (6 hours time in bed) resulted primarily in a reduction of REM sleep, interestingly during the following sleep extension REM sleep duration increased. As stated by Scorucack et al. “in the search for the mechanisms underlying the negative consequences of insufficient sleep, the implication of a REM sleep deficit should be considered”. Klerman et al. [85] have recently reported a striking increase in REM sleep during sleep extension over baseline, suggesting a rebound phenomena from a possible REM deprivation which occur in habitual sleep opportunity. A rebound in REM sleep has been also observed after space shuttle mission, during which sleep duration was approximately 6.5 hour [86]. Singh et al. [87] reported the occurrence of SOREMP (short REM latency period), an index of increased REM pressure, in a population sample of 333 adults.  Of the variables assessed (MSLT, Epworth Sleepiness Scale, and total sleep time from nocturnal polysomnography), objective sleepiness, as determined by the MSLT, was the only measure significantly associated with 2 or more SOREMPs. 

Whereas slow wave sleep might serve a need created by wake, the functional significance of REM sleep is less clear, Benington and Heller [88] have suggested a homeostatic relationship between NREM sleep and REM sleep, that REM serves a need created during NREM sleep. Data in animals [88][89] and in humans [66] have shown that the REM intervals are regulated by homeostatic rules, REM pressure accumulates during the NREM episodes, and is dissipated during the successive REM episode. A longer REM episode causes stronger dissipation of the pressure and thus requires a longer interval to accumulate pressure before the next REM episode can occur. Disruption of REM sleep, by interrupting the pressure dissipation, can impair the quality and continuity of NREM sleep, since due to REM homeostatic mechanism, interruption of REM discharge result in more frequent attempt to enter REM causing fragmentation of NREM sleep [90].

Most hypothesis concerning REM sleep have focused on the similarity between neurophysiological events occurring in REM sleep and wakefulness [91]. Broughton [92] reported that the visual evoked potentials of subjects after a forced arousal from REM sleep are similar to those of wakefulness, whereas after a forced arousal from stage 4 (SWS), subjects show slower visual evoked potentials. Langford et al. [93] showed that spontaneous arousal from sleep were more likely to occur in REM sleep than in other sleep stages. Lavie et al [94] found that subjects instructed to wake at a specified time during the night, with no aid of an external alarm clock, mainly awakened from REM sleep. As originally hypothesized by Moruzzi [95] a continuum of increasing arousal levels exists through NREM sleep, REM sleep and wake.

A possible REM sleep function is to provide a periodic activation of the brain during sleep without inducing wakefulness or disturbing the continuity of sleep [96]. Klemm [97] has proposed that the brain uses REM to help wake itself up after it has had a sufficient amount of sleep. Similarly, Horne [98] has suggested that REM seems more likely to prepare for ensuing wakefulness than provides recovery from prior wakefulness, as happens with “deeper” NREM.

The correct phase relationship of the sleep period with the circadian pacemaker is also an important factor to guarantee adequate restorative sleep duration and sleep continuity thus providing the necessary background for a good night sleep. Due to the fact that REM sleep is controlled by the circadian clock [99][100][101], it can provide a window-like mechanism that define the termination of the sleep period when there is still the necessity to complete sleep process (not only wake related homeostasis), and to meet the circadian end of sleep timing. Furthermore, cyclic occurrence of REM sleep could contribute to sleep continuity, while periodically activating the brain and prepare it to the return to consciousness.

Mechanisms that underlie REM occurrence during the night appear thus functional to guarantee sleep continuity and duration.    

Arousal processes “monitored” by the frequency of REMs can finally further delineate the end of sleep and the awakening, consistently with the reported evidence that awakenings occur much more likely out of REM period characterized by high frequency of REMs [60][56]. REM sleep can thus be necessary to guarantee sleep continuity, prepare the brain to the wake period, and define the natural end of the sleep process.

There are two categories of sleep quality markers: a first type that refers to a specific feature of sleep (as sleep onset latency or Slow Wave Sleep); a second type that derives from a combination of more features (as sleep efficiency or Total Sleep Time). Such categorization does not clearly emerge in the present form of the manuscript. Precisely, because the concept of sleep quality is multifactorial, the indicators of the first type have proved unreliable. On the contrary, the indicators of the second type are more effective from an operational point of view. Reconsidering the possible role of REM sleep as parameter to assess sleep quality could seem unsuccessful if we consider it as an indicator of the first type, unless this reassessment is inserted within a more articulated framework.

The framework that link REM sleep to the whole sleep process is introduced and discussed in different part of the paper (i.e., sleep efficiency), lines 245-253; lines 364-374

As previously discussed in the present paper, among the different PSG variables suggested as objective indicators of sleep quality, short total duration of sleep and decreased sleep efficiency appear to be those more consistently reported as indicators of low sleep quality. Considering that short sleep duration implicates a significant reduction of REM sleep, since REM sleep mostly occur during the second half of the night, this sleep component can be critical in determining sleep quality, alteration of the systems controlling REM sleep can also produce sleep fragmentation contributing to a decreased sleep efficiency. Furthermore, use of artificial light at evening and early night hours can delay and disturb circadian rhythms, especially affecting REM sleep

 Whereas slow wave sleep might serve a need created by wake, the functional significance of REM sleep is less clear, Benington and Heller [88] have suggested a homeostatic relationship between NREM sleep and REM sleep, that REM serves a need created during NREM sleep. Data in animals [88][89] and in humans [66] have shown that the REM intervals are regulated by homeostatic rules, REM pressure accumulates during the NREM episodes, and is dissipated during the successive REM episode. A longer REM episode causes stronger dissipation of the pressure and thus requires a longer interval to accumulate pressure before the next REM episode can occur. Disruption of REM sleep, by interrupting the pressure dissipation, can impair the quality and continuity of NREM sleep, since due to REM homeostatic mechanism, interruption of REM discharge result in more frequent attempt to enter REM causing fragmentation of NREM sleep [90].

Another somewhat confused element is the lack of distinction between the macro and micro structure of sleep. It seems to me that adopting this distinction could improve the work. In this perspective, it should be very important a reference to the Cyclic Alternating Pattern (CAP), completely missing in this version.

According to reviewer’s suggestion, micro structure of sleep have been added, lines 160-169; 415-422.

Sleep micro structure can also provide information on sleep continuity and stability. Laffan et al. [44] analyzed polysomnography recordings of a sample of 5.684 participants to the “Sleep Heart Health Study”, high overall sleep stage transition rate was associated with restless and light sleep fragmentation, analysis of stage-specific transition rates showed that transitions between wake and NREM sleep were also independently associated with restless and light sleep. Further analysis of the sleep micro structure is provided by the measurement of the cyclic alternating pattern (CAP) of NREM sleep [45]. CAP rate is considered to reflect arousal level during sleep. Terzano et al. [46] reported that altered CAP rate in subjects with insomnia was reduced by hypnotic treatment and that the most significant correlation between sleep quality and PSG variables was found for CAP rate

PSG has been mainly used for traditional sleep scoring analysis, with few studies addressed to analysis of sleep stage transitions and of cyclic alternating pattern, that in patient populations have shown promising insights on the mechanisms of sleep organization. Future studies on sleep quality in healthy subjects should thus be extended to analysis of sleep micro structure, to better characterize those mechanisms of sleep that can result in a poor sleep quality. Also, PSG assessment of sleepiness and wakefulness with MSLT and MWT, that are used at clinical level to diagnose sleep disorders, will contribute to provide objective information on daily functioning of healthy subjects.

I do not entirely agree with the statement that the subjective measures are not able to focus on the sleep features that can guarantee sleep quality. Perhaps, the author should better explain what he means by sleep quality.

The statement has been changed to better explain the role of objective mesures, line 35-40

Subjective estimate of sleep latency, sleep duration, presence of “insomnia”, use of sleeping drugs and daytime functioning constitute the general background for the assessment of sleep quality, however all these measures are not able to focus on the specific sleep components that can lie behind sleep quality and furthermore do not give cues on biological mechanisms that might be responsible for altering sleep process and disturb sleep continuity.

Minor concerns

When Author quotes works on sleep ad libitum, he should comment this taking into account the spontaneous trend of biological clock to run longer than 24 hours.

The statement is strictly related to citation of the seminal work of Webb

At page 2, line 96, it is present one repetition mistake: “low subjective sleep quality”

Yes, the mistake has been corrected, thanks.

At page 3, line 152, a mark should be moved.

The typo has been corrected, thanks

Out of 76 bibliographic references, only 5 (6.5%) are from the last two years.

The following recent references have been added:

Ocampo-Garcés. A.; Bassi, A.; Brunetti, E.; Estrada, J.; Vivaldi, E.A. REM sleep-dependent short-term and long-term hourglass processes in the ultradian organization and recovery of REM sleep in the rat. Sleep. 2020, 12, 43(8):zsaa023. doi: 10.1093/sleep/zsaa023.

Barrea, L.; Pugliese, G.; Framondi, L.; Di Matteo, R.; Laudisio, D; Savastano, S.; Colao, A.; Muscogiuri, G. Does Sars-Cov-2 threaten our dreams? Effect of quarantine on sleep quality and body mass index. J Transl. Med. 2020 18, 318.  doi: 10.1186/s12967-020-02465-y.

Skorucak, J.; Arbon, E.L.; Dijk, D.J.; Achermann, P. Response to chronic sleep restriction, extension, and subsequent total sleep deprivation in humans: adaptation or preserved sleep homeostasis? Sleep. 2018, 41, 1-17 . doi: 10.1093/sleep/zsy078.

Klerman, E.B.; Barbato, G.; Czeisler, C.A.: Wehr, T.A. Can people sleep too much? effects of extended sleep opportunity on sleep duration and timing. Front. Physiol. 2021,  doi: 10.3389/fphys.2021.792942

In the references section, there are some words in bold style that should be changed (for example, in the reference number 32 the word “spectral”).

Yes, several typos have been corrected, thanks.

Round 2

Reviewer 1 Report

The manuscript got improved based on the review comments! 

Reviewer 2 Report

Thanks for Good response to reviewers and excellence restructering

Reviewer 3 Report

The revised version of the manuscript is improved